# *SHROOM3* Deficiency Aggravates Adriamycin-Induced Nephropathy Accompanied by Focal Adhesion Disassembly and Stress Fiber Disorganization

**DOI:** 10.3390/cells14120895

**Published:** 2025-06-13

**Authors:** Li-Nan Xu, Ying-Ying Sun, Yan-Feng Tan, Xin-Yue Zhou, Tian-Chao Xiang, Ye Fang, Fei Li, Qian Shen, Hong Xu, Jia Rao

**Affiliations:** 1Department of Nephrology, Children’s Hospital of Fudan University, National Pediatric Medical Center of China, Shanghai 201100, China; 22111240022@m.fudan.edu.cn (L.-N.X.); sunyingyingsyy@163.com (Y.-Y.S.); 18964987829@163.com (X.-Y.Z.); fangyenzb@163.com (Y.F.); shenqian@shmu.edu.cn (Q.S.); 2Shanghai Key Lab of Birth Defect, Children’s Hospital of Fudan University, Shanghai 201100, China; yf_tan@fudan.edu.cn (Y.-F.T.);; 3Department of Developmental and Behavioral Pediatric & Child Primary Care, Ministry of Education-Shanghai Key Laboratory of Children’s Environmental Health, Xin Hua Hospital Affiliated Shanghai Jiao Tong University School of Medicine, Shanghai 200092, China; feili@shsmu.edu.cn; 4National Key Laboratory of Kidney Diseases, Beijing 100039, China; 5Department of Pediatric Nephrology, Rheumatology and Immunology, Xinhua Hospital Affiliated to Shanghai Jiao Tong University School of Medicine, Shanghai 200092, China

**Keywords:** *SHROOM3*, podocyte, ADR-induced nephropathy

## Abstract

*SHROOM3* encodes an actin-binding protein involved in kidney development and has been associated with chronic kidney disease through genome-wide association studies. However, its regulatory role in proteinuric kidney diseases and its mechanistic contributions to podocyte homeostasis remain poorly defined. Here, we analyzed single-cell transcriptomic datasets and the Nephroseq database to delineate *SHROOM3* expression patterns in proteinuric kidney diseases. Using podocyte-specific *SHROOM3* knockout mice and an Adriamycin (ADR)-induced nephropathy mouse model, we demonstrated that glomerular *SHROOM3*, specifically in podocytes, was upregulated following ADR treatment during the acute injury phase but downregulated in chronic kidney disease. Clinically, the glomerular *SHROOM3* expression positively correlated with glomerular filtration rates in focal segmental glomerulosclerosis patients. Genetic ablation of *SHROOM3* in podocytes exacerbated ADR-induced proteinuria, diminished podocyte markers (nephrin, podocin, and WT1), and accelerated glomerulosclerosis. In vitro, *SHROOM3* deficiency impaired podocyte size and adhesion, concomitant with the downregulation of focal adhesion molecules (talin1, vinculin, and paxillin) and stress fiber regulators (synaptopodin and RhoA), as well as calpain activation and RhoA inactivation. Our findings reveal a critical role for *SHROOM3* in maintaining podocyte integrity and suggest its therapeutic potential in mitigating proteinuric kidney disease progression.

## 1. Introduction

Nephrotic syndrome (NS), a clinical manifestation of glomerular diseases characterized by podocyte abnormalities, represents an important contributor to chronic kidney disease (CKD) particularly in steroid-resistant forms [1]. Genetic and experimental studies underscore the critical role of podocyte homeostasis, which relies on a precisely regulated actin cytoskeleton network that maintains structural integrity and glomerular filtration function. Beyond its structural role, the podocyte cytoskeleton serves as a signaling nexus, coordinating molecular cascades that govern cellular processes and adaptive responses [2].

*SHROOM3* encodes an evolutionarily conserved actin-binding protein that regulates cell morphology and tissue architecture through cytoskeletal interactions [3,4,5,6]. Initially identified in neural tube defects, where its deficiency causes embryonic lethality via exencephaly in mouse models [3], *SHROOM3* was later implicated in renal pathology when heterozygous null mice developed glomerulosclerosis and foot process effacement by one year of age [7]. These defects were attributed to ROCK/nonmuscle myosin II axis dysregulation, disrupting actin dynamics [7]. Genome-wide association studies (GWAS) further linked *SHROOM3* locus variants to CKD, reduced the estimated glomerular filtration rate (eGFR), and albuminuria [8,9,10], suggesting its regulatory role in kidney function. While emerging evidence points to a protective function in focal segmental glomerulosclerosis (FSGS) [11], the mechanistic basis of *SHROOM3* in podocyte pathophysiology remains unclear.

Given the established involvement of *SHROOM3* in glomerulosclerosis and the podocytopathies, we comprehensively investigated its role in podocytes under both physiological and pathological conditions. Using podocyte-specific *Shroom3* knockout mice and transcriptomic approaches, we assessed *SHROOM3* function in vivo and in vitro, focusing on Adriamycin (ADR)-induced nephropathy, a validated FSGS model characterized by progressive podocyte loss and glomerulosclerosis.

## 2. Materials and Methods

### 2.1. Data Sources and Analysis Methods

Single-nucleus RNA sequencing (snRNA-seq) data from healthy adult human and mouse kidneys were accessed through the Kidney Interactive Transcript (KIT) database (https://humphreyslab.com/SingleCell/, accessed on 28 February 2025). Human kidney transcriptomic profiles were obtained from the “Healthy Adult Human Kidney” dataset [12], while mouse kidney transcriptional data were derived from the “Healthy Mouse dataset” [13]. These datasets were analyzed using the built-in visualization tools provided by the KIT platform.

For the assessment of *SHROOM3* expression in proteinuric kidney diseases, we analyzed four independent cohorts from the publicly available Nephroseq database (https://www.nephroseq.org/resource/login.html, accessed on 28 February 2025). Cohort 1 (“Nakagawa CKD Kidney” dataset) comprised transcriptomic data from renal biopsy specimens of 53 chronic kidney disease patients and 8 healthy controls [14]. Cohort 2 (“Mariani Nephrotic Syndrome Glom” dataset) consisted of glomerular transcriptomes from patients with nephrotic syndrome, including 93 FSGS and 89 minimal change disease (MCD) patients [15]. Cohort 3 (“Mariani Nephrotic Syndrome TubInt2” dataset) included glomerular transcriptomic profiles from 15 FSGS and 20 MCD patients [15]. Cohort 4 (“Hodgin FSGS Glom” dataset) comprised 8 FSGS patients and 9 healthy controls [16]. Clinical parameters (proteinuria and eGFR) and *SHROOM3* expression levels were extracted from all four cohorts.

To investigate cell-specific expression of *SHROOM3* in proteinuric kidney disease, we analyzed single-cell RNA sequencing (scRNA-seq) data from the glomeruli of Adriamycin (ADR)-induced nephropathy mice [17] obtained from Gene Expression Omnibus (GEO) (accession number GSE146912). Glomeruli were isolated using magnetic bead enrichment, yielding populations predominantly composed of glomerular cells with minor populations of tubular and immune cells. For our analysis of glomerular *SHROOM3* expression, we focused exclusively on glomerular cell populations, including podocytes, mesangial cells and endothelial cells. Quality control measures excluded cells that did not meet the following criteria: (1) 500–10,000 expressed genes; (2) 1000–100,000 unique molecular identifiers (UMIs); (3) <5% mitochondrial genes; and (4) <5% hemoglobin genes. Data were processed and analyzed using the Seurat package (version 4.1.1) in R.

### 2.2. Animal Models of ADR-Induced Nephropathy

The Adriamycin (ADR)-induced nephropathy model was established as a murine model of human FSGS as previously described [18]. Eight-week-old male mice (with an average body weight of 22−25 g) received a single tail vein injection of either ADR (17 mg/kg; HY-15142, MCE, USA) or an equivalent volume of saline. All experiments included at least three independent biological replicates with technical triplicates to ensure reproducibility. No data points were excluded except for pre-defined outliers exceeding mean ± 3SD. Mice were monitored for a period of 10 weeks post-injection. Body weight and urine samples were collected at designated timepoints (2, 6, 8 and 10 weeks after ADR injection), while renal tissues were harvested at week 2 and 10 post-injection for subsequent analyses. Mice were anesthetized with isoflurane (5% for induction and 2% for maintenance) followed by cervical dislocation. Tissues were harvested within 2 min post-euthanasia to minimize degradation.

### 2.3. Podocyte-Specific Shroom3 Knockout Mice

All mice were housed in a specific-pathogen-free facility at temperatures of 21–24 °C with 40–70% humidity on a 12 h light/12 h dark cycle and provided with food and water ad libitum. Wild-type male C57BL/6J mice (purchased from Cyagen, Shanghai, China) were bred in the animal facility of the Children’s Hospital of Fudan University. All animals, including both experimental and control mice, were co-housed with free access to water and standard animal chow. The ARRIVE guidelines 2.0 were followed for all animal experiments [19].

To generate podocyte-specific *SHROOM3* knockout mice on the C57BL/6J background, breeding pairs of *Shroom3*^flox/+^ mice were generated by inserting loxP cassettes into the 4th and 5th introns of the *Shroom3* genomic locus using embryonic-stem-cell targeting (Cyagen, Shanghai, China). Cre-mediated deletion of exon 5 results in a truncated transcript with a premature stop codon, leading to transcript degradation via nonsense-mediated decay and subsequent absence of protein production [20]. The tdTomato reporter mice (Strain #007909, JAX) contain a loxP-flanked STOP cassette preventing transcription of a CAG-promoter-driven tdTomato, which is inserted into the Gt(ROSA)26Sor locus. Breeding *Nphs2*-Cre (Strain #008205, JAX) mice with tdTomato mice generated podocyte-specific tdTomato expression. To create the podocyte-conditional *Shroom3* knockout (*Shroom3*-PKO), *Shroom3*^flox/+^ mice were crossed with *Nphs2*-Cre mice and tdTomato mice for at least 3 generations. Littermates with the *Shroom3*^flox/flox^ genotype served as controls. The genotype was verified by PCR of tail DNA using specific primers (Appendix A).

### 2.4. Proteinuria and Renal Function Assessment

Urine samples were collected weekly from mice after injection with Adriamycin or saline and analyzed for the urine-albumin-to-creatinine ratio (UACR). After collection, samples were centrifuged at 4000× *g* for 15 min at 4 °C. The supernatant was collected for subsequent analysis. As described previously [18], the urine albumin concentration was determined by loading 5 µL of a urine sample per lane on a sodium dodecyl sulfate-polyacrylamide gel for electrophoresis, followed by Coomassie blue staining, with bovine serum albumin (BSA, 0.4 μg and 0.2 μg) serving as quantification standards. The urine creatinine concentration was measured using a creatinine assay kit (Nanjing Jiancheng Bioengineering Institute, Nanjing, China) according to the manufacturer’s instructions. The UACR was calculated by dividing the urine albumin concentration by the creatinine concentration.

Blood samples were collected from 3-month-old mice via retro-orbital bleeding and were allowed to clot at room temperature for 30 min and then centrifuged at 1,200 × g for 15 min at 4°C to obtain serum. Serum creatinine levels were measured using a creatinine assay kit (Nanjing Jiancheng Bioengineering Institute, China) following the manufacturer’s protocol. Serum creatinine concentrations served as indicators of the glomerular filtration rate and overall kidney function.

### 2.5. Histology and Immunostaining

Mouse kidneys were harvested and promptly decapsulated. For molecular analyses, tissue samples were snap-frozen for subsequent protein and RNA extraction. For histological examination, longitudinally bisected kidneys were fixed in 4% paraformaldehyde (PFA) in phosphate-buffered saline (PBS) at 4 °C overnight, followed by paraffin embedding. Paraffin sections (4 μm) were stained with periodic acid–Schiff (PAS) and Masson’s trichrome following standard protocols.

### 2.6. Isolation of Glomeruli and Tubules of Kidney Cortex

Mouse glomeruli were isolated using a sequential sieving method as previously described (*n* ≥ 3) [21]. Briefly, harvested kidneys were decapsulated and minced into small fragments, followed by enzymatic digestion with collagenase A (1 mg/mL; Roche, BSL, Switzerland) at 37 °C for 20 min. The digested tissue was filtered through a series of cell strainers (100, 70 and 40 μm) using ice-cold Hanks’ Balanced Salt Solution (HBSS; Gibco, Carlsbad, CA, USA, #14185052). Glomerular enrichment was achieved through differential adhesion, allowing tubular fragments to adhere while glomeruli remained in suspension. The purity of isolated glomeruli was microscopically confirmed to be 90–95% (Appendix A). The purified glomeruli and adhered tubular fragments were subsequently processed for RNA extraction and further analyses.

### 2.7. Cell Culture and Treatment

Conditionally immortalized human podocytes (a gift from Dr. Moin Saleem, University of Bristol, Southmead Hospital, Bristol, UK) were maintained in proliferation medium containing RPMI-1640 supplemented with 10% fetal bovine serum (FBS), 1% insulin-transferrin-selenium (ITS) and 1 × penicillin-streptomycin (all from Gibco) at 33 °C with 5% CO_2_. For *SHROOM3* knockdown, proliferating podocytes were transduced with lentiviral particles carrying either *SHROOM3*-specific shRNA (Appendix A) or non-targeting control shRNA. Stable cell lines were selected using puromycin (5 μg/mL) for 7 days. Prior to thermal shifting, culture plates were coated with collagen type I (10 mg/mL) for 24 h to enhance cell attachment. Podocytes were then thermoshifted to 37 °C and cultured in differentiation medium (without ITS) for 10–14 days to induce terminal differentiation.

### 2.8. Quantification of Podocyte Size, Adhesion and Apoptosis

Podocyte size was assessed via forward scatter (FSC) measurements using flow cytometry (BD FACSCanto II) [22]. Cell adhesion assays were performed in collagen type I-coated 96-well plates, as previously described [23]. Following washing, adherent cells were manually counted under a light microscope at 100 × magnification in three random fields per well across five independent wells. For apoptosis analysis, cells were harvested and stained with Annexin V-FITC/propidium iodide (PI) using an Annexin V-FITC Apoptosis Detection Kit (BD Biosciences) according to the manufacturer’s protocol [24]. Briefly, cells were washed twice with cold PBS and resuspended in 1 × binding buffer to achieve a concentration of 1 × 10⁶ cells/mL. The cell suspension (100 μL) was incubated with 5 μL of Annexin V-FITC and 5 μL of PI for 15 min at room temperature in the dark. Following the addition of 400 μL of binding buffer, the samples were analyzed using a flow cytometer (BD FACSCanto II). Early apoptotic (Annexin V-positive/PI-negative) and late apoptotic (Annexin V-positive/PI-positive) cells were quantified and expressed as percentages of the total cell population.

### 2.9. Calpain Activity Assay

Calpain activity was measured using a Calpain Activity Assay Kit (Abcam, Cambridge, UK, #ab65308). Differentiated podocytes (cultured for 10–14 days) were lysed in extraction buffer supplemented with 1% Triton X-100 to evaluate membrane-associated calpain activity. Protein concentration was determined by BCA assay to ensure equal loading. Cell lysates (100 μg protein per well) were incubated with calpain substrate in a 96-well plate. Active calpain and calpain inhibitors were used as positive and negative controls, respectively, to validate assay specificity. Following a 1-h incubation at 37 °C in the dark, fluorescence was measured at excitation/emission wavelengths of 400/505 nm using a microplate reader.

### 2.10. RhoA Activity Assay

Active RhoA levels were measured using a G-LISA-RhoA Activation Assay Kit (Cytoskeleton Inc., Denver, CO, USA, Cat. #BK124) according to the manufacturer’s instructions. Control and *SHROOM3* KD podocytes were lysed in ice-cold lysis buffer and equal amounts of protein (30 μg) were loaded onto pre-treated G-LISA plates. Following antibody incubations and signal development, absorbance was measured at 490 nm. Results are expressed as fold change relative to the control group from three independent experiments to ensure reproducibility.

### 2.11. Real-Time Quantitative PCR (RT-qPCR)

Renal cortex and isolated glomeruli were processed for RNA extraction. Total RNA was extracted using TRIzol reagent (Invitrogen, Carlsbad, CA, USA) followed by phenol-chloroform purification. RNA concentration and purity were measured using a NanoDrop spectrophotometer (Thermo Fisher Scientific, Waltham, MA, USA). Only samples with 260/280 ratios between 1.8–2.0 and 260/230 ratios > 1.8 were used for further analysis. For all samples, 1 μg of total RNA was reverse transcribed using PrimeScript RT Master Mix (Takara, Tokyo, Japan) under identical conditions. Quantitative PCR was performed using SYBR Green SuperMix (Vazyme, Nanjing, China) on a QuantStudio 3 thermal cycler. Gene expression was normalized to glyceraldehyde-3-phosphate dehydrogenase (*GAPDH*), which demonstrated consistent expression across all experimental conditions with CT values not differing by more than 1 cycle. Gene expression was quantified using the 2^−ΔΔCT^ method, where ΔCT = CT (target gene) − CT (reference gene). ΔΔCT = ΔCT (experimental sample) − ΔCT (control sample). Fold change = 2^−ΔΔCT^. Primer sequences used for all RT-PCR experiments are listed in Appendix A.

### 2.12. Immunofluorescence Analysis

Cultured podocytes were fixed with 4% paraformaldehyde for 10 min at room temperature, permeabilized with 0.3% Triton X-100 in PBS for 5 min and blocked with 5% BSA for 1 h. Focal adhesions were visualized using anti-paxillin antibody (BD Biosciences, San Jose, CA, USA, #610619, 1:200) and F-actin stress fibers were labeled with Alexa Fluor-conjugated phalloidin (Invitrogen). After overnight incubation with the primary antibody at 4 °C, cells were incubated with fluorophore-conjugated secondary antibodies for 1 h at room temperature. Nuclei were counterstained with DAPI. Images were captured using a confocal laser scanning microscope (Leica TCS SP8) with consistent acquisition settings across all experimental conditions.

Fluorescence in situ hybridization (FISH) was performed on 10 μm fresh-frozen kidney sections from newborn mice. Tissues were fixed with 4% paraformaldehyde, permeabilized with 0.3% Triton X-100 and hybridized overnight at 40 °C with green fluorescent-labeled *Shroom3*-specific oligonucleotide probes following the manufacturer’s protocol (Advanced Cell Diagnostics, Newark, CA, USA). Native tdTomato fluorescence was preserved to identify Cre-expressing podocytes.

### 2.13. Western Blot Analysis

Cell lysates were collected and total protein levels were quantified by a Bradford assay. Equal amounts of protein were separated by SDS-PAGE and transferred to PVDF membranes. The following primary antibodies were used: *SHROOM3* (Sigma-Aldrich, St. Louis, MO, USA, #SAB3500818, 1:1000), Calpain1 (Abcam, #ab108400, 1:3000), Synaptopodin (Santa Cruz, Shanghai China, #sc-515842, 1:1000), RhoA (Santa Cruz, #sc418, 1:500), Paxillin (BD Biosciences, #610619, 1:4000), Talin1 (Abcam, #ab108480, 1:1000) and Vinculin (Sigma, #V9131, 1:5000). After incubation with HRP-conjugated secondary antibodies (Cell Signaling Technology, Danvers, MA, USA), protein bands were visualized using an Ultra High Sensitivity ECL Kit (MCE) and imaged using a BioRad Gel Doc XR system. Densitometric analysis was performed using ImageJ v1.8.0.345 software, with protein expression levels normalized to GAPDH in the same sample.

### 2.14. RNA Sequencing and Analysis

Total RNA was extracted from differentiated *SHROOM3*-knockdown podocytes and control podocytes for transcriptomic profiling. Libraries were prepared using a NEBNext Ultra II RNA Directional Kit and sequenced on an Illumina NovaSeq 6000 platform. Raw reads were demultiplexed using bcl2fastq2, subjected to quality control assessment (FastQC) and trimmed (Trimmomatic) to remove adapters and low-quality bases. Processed reads were aligned to the GRCh38 reference genome using HISAT2 and gene counts were generated using HTSeq. Differentially expressed genes (DEGs) were identified using DESeq2, with significance thresholds set at Log_2_|FC| > 1 and *p* < 0.05. Gene Set Enrichment Analysis (GSEA) was performed to identify differentially enriched biological pathways (*p* < 0.05, FDR < 0.05) between the two experimental groups using the “clusterProfiler” package in R software. The RNA sequencing data generated in this study have been deposited in the Gene Expression Omnibus (GEO) database (GSE294734).

### 2.15. Statistical Analyses

All data analyses were performed using GraphPad Prism 9 and R 4.2.2 software. Comparisons between multiple groups were made using one-way analysis of variance (ANOVA), followed by the Student–Newman–Keuls post-hoc test for pairwise comparisons. Comparisons between two groups were made using Student’s *t*-test for normally distributed data. For data that failed normality testing, a Mann–Whitney U test was applied to compare groups. Results are presented as mean ± standard error of the mean (SEM) unless otherwise specified. A *p*-value < 0.05 was considered statistically significant.

## 3. Results

### 3.1. Dynamic Regulation and Clinical Relevance of SHROOM3 Expression in Proteinuric Kidney Diseases

To investigate the potential role of *SHROOM3* in nephropathy, we initially assessed its distribution across kidney cell types using the Kidney Interactive Transcriptomics (KIT) database. As demonstrated in Appendix A and Figure 1A, *SHROOM3* is broadly expressed throughout the healthy adult human kidney, with predominant expression in podocytes. Similar cell-specific expression patterns was observed in adult mouse kidneys (Appendix A).

To understand how *SHROOM3* responds during kidney injury, we analyzed a single-cell RNA sequencing (scRNA-seq) dataset from mice with ADR-induced nephropathy. This confirmed the predominant expression of *SHROOM3* in podocytes and revealed elevated expression two weeks after ADR injection compared to saline-injected controls (Figure 1B and Appendix A), suggesting *SHROOM3* plays a pivotal role during early proteinuric kidney injury. Given the podocyte-predominant expression of *SHROOM3* and its association with multiple CKD-related GWAS studies, we next evaluated *SHROOM3* expression in patients with proteinuric kidney disease. Analysis of transcript datasets from Nephroseq demonstrated significantly higher *SHROOM3* abundance in kidney biopsies from CKD patients compared with normal controls (Figure 1C).

In a cohort of patients with nephrotic syndrome, glomerular *SHROOM3* expression was markedly lower in patients with eGFR < 60 mL/min/1.73 m^2^ compared to those with eGFR ≥ 60 mL/min/1.73 m^2^ (Appendix A) and significantly correlated with eGFR levels (Appendix A). In contrast, tubulointersititial *SHROOM3* expression showed no significant correlation with eGFR levels (Appendix A). When analyzing disease-specific correlations, we observed a strong positive relationship between glomerular *SHROOM3* expression and eGFR specifically in patients with FSGS (Figure 1D,E), whereas no significant correlation appeared in minimal change disease (MCD) (Appendix A). Notably, glomerular *SHROOM3* expression also positively correlated with proteinuria severity in the FSGS cohort. (Figure 1F,G). These clinical correlations collectively suggest the involvement of *SHROOM3* in the pathophysiology of proteinuric kidney diseases, particularly FSGS.

### 3.2. Induction of Shroom3 Expression in Podocytes Following Glomerular Injury

To further explore how *Shroom3* expression changes during progressive glomerular damage, we employed the ADR-induced nephropathy mouse model (Figure 2A), a well-established experimental system that recapitulates podocyte injury and proteinuria. As expected, ADR treatment induced progressive proteinuria, with significant increases in UACR from week 2 to week 10 post-injection (Figure 2B). Histological examination using periodic acid–Schiff (PAS) staining revealed glomerular adhesion 2 weeks after ADR injection, while, by 10 weeks, significant proliferation of glomerular mesangial cells and marked matrix deposition were observed (Figure 2C).

Quantitative analysis revealed a significant upregulation of *Shroom3* expression in both kidney cortex and isolated glomeruli at week 2 post-ADR injection compared to saline-treated controls (Figure 2D,E). Interestingly, by week 10, as chronic kidney damage progressed, *Shroom3* expression in both renal cortex and glomeruli declined (Figure 2D,E). In contrast, its expression in renal tubules continued to rise throughout the disease course (Figure 2F). These findings highlight compartment-specific and temporally regulated expression patterns of *Shroom3* during kidney injury progression. Notably, the early elevation of *Shroom3* specifically in glomeruli during the initial phase of ADR-induced podocyte injury suggests it functions as an early adaptive response to glomerular damage.

### 3.3. Establishment of Podocyte-Specific Ablation of Shroom3 Mouse Model

To investigate the functional significance of *Shroom3* in podocytes, we generated podocyte-specific *Shroom3*-knockout mice using a Cre-LoxP gene targeting approach (Figure 3A). We crossed *Nphs2*-Cre mice, which express Cre recombinase specifically in podocytes, with a *Shroom3* floxed line (*Shroom3*^flox/+^) containing loxP sites flanking exon 5 of the *Shroom3* gene. To validate the efficiency of Cre-mediated recombination in podocytes, we incorporated a fluorescent reporter (tdTomato) into the breeding strategy. After at least three generations of breeding, we obtained *Shroom3*^flox/flox^*Nphs2*^Cre^-tdTomoto (*Shroom3*-PKO) mice, with *Shroom3*^flox/flox^ mice serving as controls (Figure 3B).

Genotyping confirmed the presence of the floxed alleles, Cre transgene and tdTomato reporter in the *Shroom3*-PKO mice (Figure 3B). Quantitative RT-PCR analysis demonstrated an approximately 80% reduction in *Shroom3* mRNA expression in isolated glomeruli from *Shroom3*-PKO mice compared to controls (Figure 3C). The remaining expression likely originated from non-podocyte glomerular cells, such as mesangial cells, which also express *Shroom3*. FISH analysis revealed significantly diminished *Shroom3* RNA signals in tdTomato-positive glomeruli of *Shroom3*-PKO mice, where tdTomato expression indicated active Cre recombinase activity (Appendix A). In contrast, control glomeruli exhibited normal *Shroom3* expression without tdTomato fluorescence. Phenotypically, *Shroom3*-PKO mice were viable, fertile and exhibited no overt abnormalities (Figure 3D). At 3 months of age, no significant differences were observed in body weight, kidney/body weight ratios, UACR, or serum creatinine between *Shroom3*-PKO mice and control littermates (Appendix A). These observations indicate that podocyte *Shroom3* is dispensable for maintaining normal glomerular filtration barrier function under physiological conditions, suggesting its role may become more critical under pathological stress.

### 3.4. Podocyte-Specific Ablation of Shroom3 Aggravates ADR-Induced Nephropathy

The absence of baseline abnormalities in *Shroom3*-PKO mice prompted us to investigate *Shroom3*’s role under pathological conditions. We challenged *Shroom3*-PKO mice and control littermates with intravenous ADR administration and collected urine samples at weeks 1–2 and week 10 post-injection to assess disease progression. Coomassie blue staining of urine samples revealed elevated albumin levels in *Shroom3*-PKO mice compared to controls following ADR administration (Figure 4A). Quantitative analysis confirmed significantly higher UACR in *Shroom3*-PKO mice at weeks 1, 2 and 10 compared with control littermates (Figure 4B,C), indicating more severe and persistent proteinuria in the absence of podocyte *Shroom3*.

To further assess podocyte integrity, we analyzed the expression of podocyte-specific markers using RT-PCR. At week 2 post-ADR injection, *Shroom3* deficiency in podocytes led to reduced mRNA expression of nephrin (*Nphs1*), podocin (*Nphs2*) and the Wilms tumor protein (*Wt1*) compared to controls (Figure 4D). By week 10, only podocin mRNA levels remained significantly different between the two groups (Figure 4E), possibly reflecting the progressive nature of podocyte injury and adaptive responses in both genotypes over time.

Histological evaluation using PAS and Masson’s trichrome staining revealed more severe renal pathology in *Shroom3*-PKO mice, characterized by increased collagen and fibrin deposition (Figure 4F,G). Collectively, these findings demonstrate that podocyte-specific deletion of *Shroom3* exacerbates ADR-induced nephropathy, highlighting its protective role against podocyte injury.

### 3.5. SHROOM3 Deficiency Impairs Podocyte Morphology and Function In Vitro

To elucidate the cellular mechanisms by which *SHROOM3* deficiency enhances susceptibility to podocyte injury, we established an in vitro model using cultured human podocytes with stable *SHROOM3* knockdown via shRNA (Appendix A). Immunofluorescence (IF) staining revealed a significant reduction in podocyte size following *SHROOM3* knockdown (Figure 5A). Flow cytometric forward scatter (FSC) measurements confirmed the significant decrease in cell size in *SHROOM3*-deficient podocytes (Figure 5B).

Functional assessment using cell adhesion assays demonstrated that *SHROOM3* knockdown significantly reduced the number of adherent podocytes (Figure 5C), suggesting impaired cell–matrix interactions. Furthermore, *SHROOM3* deletion promoted apoptosis (Figure 5D,E), as evidenced by increased Annexin V staining and compromised the proliferative capacity of podocytes in their proliferative state (Figure 5F). These in vitro findings complement our in vivo observations and demonstrate that *SHROOM3* plays a critical role in maintaining podocyte morphology, adhesion and survival, highlighting its importance in podocyte homeostasis that is essential for glomerular filtration barrier integrity.

### 3.6. mRNA Profiling Analysis Reveals Calpain Activation in SHROOM3-Deficient Podocytes

To comprehensively identify the molecular pathways affected by *SHROOM3* deficiency, we performed RNA sequencing (RNA-seq) on *SHROOM3*-knockdown and control human podocytes. Principal component analysis (PCA) and sample correlation analysis demonstrated tight clustering within each group and clear separation between groups, indicating good experimental consistency (Appendix A). Differential expression analysis identified 1462 DEGs with statistical significance (*p* < 0.05 and Log_2_|FC| ≥ 1), including 1055 upregulated and 407 downregulated genes in *SHROOM3*-deficient podocytes (Figure 6A).

Hierarchical clustering of RNA-seq data revealed distinct expression patterns of known podocytopathic genes between *SHROOM3*-knockdown and control cells (Figure 6B). Among the most significantly altered genes, we identified marked downregulation of complement factor H (*CFH*), a key inhibitor of the alternative complement pathway [25], as well as *ITGA3* and *ITGB4*, which encode integrin subunits critical for podocyte–glomerular-basement-membrane adhesion. Gene Set Enrichment Analysis (GSEA) identified significant alterations in multiple biological pathways in *SHROOM3*-deficient podocytes (Figure 6C). Notably, processes related to cytoskeletal organization, cell–matrix adhesion and cell polarity were downregulated, while inflammatory and immune response pathways were upregulated. These findings highlight the pleiotropic roles of *SHROOM3* in regulating podocyte structure, adhesion and immune responses.

To validate these transcriptomic findings at the protein level, we performed immunofluorescence staining for F-actin (using phalloidin) and paxillin in *SHROOM3* knockdown podocytes. We observed disorganized actin stress fibers and reduced paxillin staining (Figure 6D), confirming cytoskeletal and focal adhesion defects. Western blot analysis confirmed decreased expression of key focal adhesion proteins, including talin1, vinculin and paxillin in *SHROOM3*-deficient podocytes compared to controls (Figure 6E,F). Given that focal adhesion molecules are susceptible to calcium-dependent degradation by calpain, we investigated whether *SHROOM3* deficiency facilitated calpain activation. In *SHROOM3*-deficient podocytes, we detected increased protein expression of Calpain 1 (µ-calpain) and enhanced calpain enzymatic activity (Figure 6G–I), suggesting dysregulated proteolysis.

We additionally examined the RhoA signaling pathway, which is critical for stress fiber formation and focal adhesion maturation. *SHROOM3*-deficient podocytes exhibited reduced RhoA protein expression and decreased active RhoA levels, without corresponding changes in RhoA mRNA expression (Figure 6J–M). This discrepancy suggested post-translational regulation of RhoA. Since previous studies have established that RhoA degradation is regulated through Smurf1-mediated ubiquitination, a process controlled by synaptopodin [26], we examined synaptopodin expression and found it significantly decreased in *SHROOM3*-deficient podocytes (Figure 6N,O).

Collectively, these molecular analyses suggest that *SHROOM3* deficiency impairs stress fiber assembly and organization through decreased synaptopodin-mediated RhoA signaling, while simultaneously enhancing calpain-dependent proteolysis of focal adhesion components. These molecular alterations ultimately disrupt podocyte cytoskeletal integrity and cell–matrix adhesion, rendering podocytes more susceptible to injury (Figure 7).

## 4. Discussion

Our multidisciplinary investigation elucidated the critical role of SHROOM3 in proteinuric kidney disease through integrated bioinformatics, animal models and cellular experimentation. Database analyses (KIT and Nephroseq) revealed predominant podocyte expression of SHROOM3 with dynamic regulation during renal injury. Clinical correlations demonstrated positive associations between SHROOM3 expression and both eGFR and proteinuria in FSGS patients. These findings were corroborated in our ADR nephropathy model, which showed early *Shroom3* upregulation during podocyte stress. Podocyte-specific *Shroom3* knockout (*Shroom3*-PKO) mice exhibited exacerbated ADR-induced proteinuria and accelerated glomerulosclerosis progression. The preserved baseline phenotype in unchallenged *Shroom3*-PKO mice contrasted with their heightened injury susceptibility, establishing SHROOM3 as a stress-responsive protector of podocyte integrity. This context-dependent function suggested physiological compensation mechanisms become overwhelmed under pathological stress.

Cellular characterization revealed SHROOM3 deficiency causes three cardinal defects: reduced podocyte size, impaired cell–matrix adhesion and increased apoptotic susceptibility. Transcriptomic profiling identified dysregulation of pathways critical for cytoskeletal organization and stress adaptation. Mechanistically, we discovered SHROOM3 depletion induces calpain hyperactivation and disrupts RhoA signaling through synaptopodin destabilization. The scaffolding function of SHROOM3 appears crucial for maintaining synaptopodin protection, thereby preserving RhoA-dependent stress fiber formation.

Our findings extend prior insights into SHROOM3’s role in renal pathophysiology. While earlier studies established that *Shroom3*-heterozygous mice developed age-dependent glomerulosclerosis and foot process effacement at one year of age [7] and heightened susceptibility to acute kidney injury [27], our work specifically delineated the podocyte-intrinsic function of *SHROOM3* through a conditional knockout model. This approach revealed developmental consequences of *Shroom3* ablation during late glomerular capillary loop formation [28] and its indispensable role in mitigating acute podocyte stress responses. By establishing *SHROOM3* as a cell-intrinsic guardian of podocyte homeostasis, we bridge the gap between developmental and acquired podocytopathies.

SHROOM3 maintained podocyte integrity through a multi-tiered regulatory network. Firstly, SHROOM3 deficiency triggered calpain hyperactivation, a calcium-dependent protease linked to podocyte injury [29,30]. This proteolytic surge coincided with focal adhesion disassembly and cytoskeletal fragmentation, mirroring the protective effects of calpain inhibition observed in prior studies [30]. Secondly, SHROOM3 depletion reduced synaptopodin levels and impaired RhoA signaling, which is critical for stress fiber assembly. We propose that SHROOM3 acts as a scaffold protein that helps stabilize synaptopodin, thereby protecting it from proteolytic degradation and preserving RhoA-driven actomyosin contractility. Previous studies have demonstrated that synaptopodin stability is regulated by 14-3-3 protein interactions [31] and importantly, direct binding between SHROOM3 and 14-3-3 proteins has also been reported [32]. Although the functional significance of this interaction in podocytes was not directly examined in our current study, these established protein–protein interactions provide a plausible mechanism for how SHROOM3 might contribute to synaptopodin stability. RNA-seq analysis confirmed broad dysregulation of cytoskeletal regulators, including downregulated Rho GTPase, actomyosin assembly and stress fiber organization pathways—consistent with observed structural deficits.

Notably, SHROOM3 deficiency disturbed the expression of integrins ITGA3/ITGB4 which is essential for glomerular basement membrane adhesion [33,34] and the complement regulator CFH, suggesting multifaceted protective mechanisms. These findings extend previous reports of glomerulosclerosis in *Shroom3* heterozygous mice by establishing cell-autonomous podocyte functions through conditional knockout strategies.

Several limitations of our study warrant consideration. First, our phenotypic characterization of *Shroom3*-PKO mice was confined to relatively young animals (1–12 weeks of age). The longitudinal studies to fully assess potential age-dependent manifestations of *Shroom3* deficiency are needed. The inclusion of larger experimental cohorts would also strengthen the statistical robustness of our findings regarding SHROOM3’s protective effects during early injury phases. Second, podocin mRNA reduction at 10 weeks post-ADR required clarification through podocyte quantification. More comprehensive podocyte quantification would help distinguish between transcriptional dysregulation and actual podocyte depletion as the primary mechanism underlying these observations. Third, upstream regulators of injury-induced SHROOM3 expression remained unidentified. Fourth, the interactions between SHROOM3, 14-3-3 and synaptopodin warranted direct experimental validation. Therapeutic potential of calpain inhibition in SHROOM3 deficiency merits investigations.

In conclusion, our findings establish SHROOM3 as a critical orchestrator of podocyte stress adaptation. The identified SHROOM3–calpain–RhoA nexus presented novel therapeutic targets for proteinuric diseases. Future studies should explore SHROOM3 augmentation strategies and pathway-specific interventions to preserve glomerular filtration barrier integrity.

## Figures and Tables

**Figure 1 cells-14-00895-f001:**
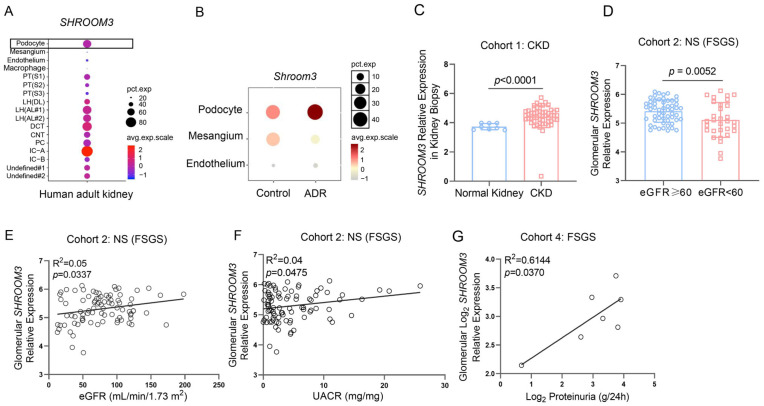
*SHROOM3* expression in kidney cell types and its clinical relevance in proteinuric diseases. (**A**) Dot plot showing *SHROOM3* expression in single-nucleus RNA sequencing (snRNA-seq) data of adult human kidney [12] using KIT database (http://humphreyslab.com/SingleCell/search.php accessed on 28 February 2025). LH: loop of Henle; DL: descending limb; AL: ascending limb; DCT: distal convoluted tubule; CNT: connecting tubule; PC: principal cells; IC: intercalated cells. (**B**) scRNA-seq analysis of glomerular-enriched cells from normal and Adriamycin (ADR)-induced nephropathy mouse kidneys (data from GSE146912). (**C**–**G**) Relationship between *SHROOM3* expression and disease groups, eGFR and UACR in the Nephroseq database. (**C**) *SHROOM3* transcript levels in biopsies of CKD patients (*n* = 53) versus normal kidneys (*n* = 8) (Cohort 1). Unpaired Student’s *t*-test. (**D**) Comparison of glomerular *SHROOM3* expression between FSGS (Cohort 2) patients with eGFR ≥ 60 (*n* = 55) and those with eGFR < 60 mL/min/1.73 m^2^ (*n* = 31). Unpaired Student’s *t*-test. (**E**,**F**) Associations of glomerular *SHROOM3* expression and eGFR levels (*n* = 86) (**E**) or UACR concentration (*n* = 91) (**F**) in FSGS (Cohort 2). Pearson correlation analysis.

**Figure 2 cells-14-00895-f002:**
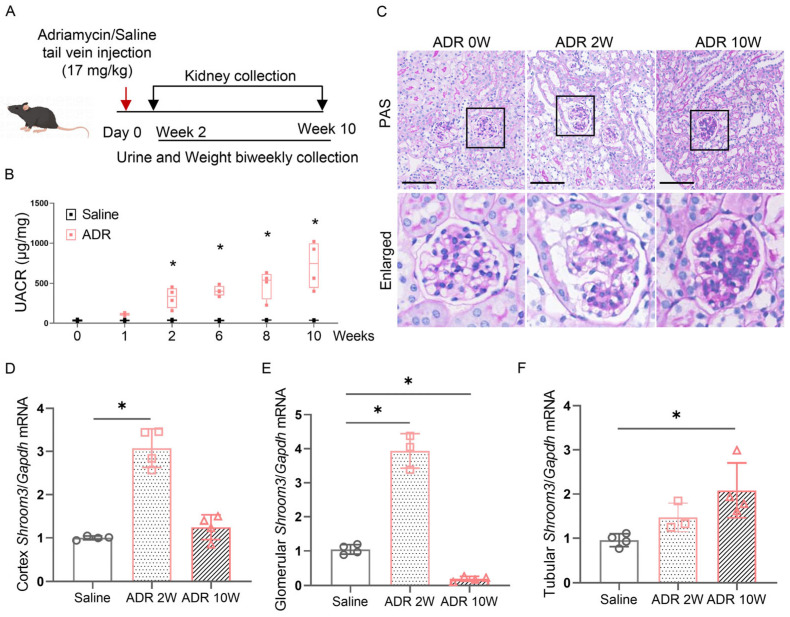
Dynamic regulation of *Shroom3* expression in ADR-induced nephropathy. (**A**) Experimental design schematic of the ADR-induced nephropathy mouse model. (**B**) Quantitative analysis of UACR in saline- and ADR-treated mice over the 10-week experimental period. Each dot represents an individual mouse datum. Unpaired Student’s *t*-test. * *p* < 0.05. (**C**) Representative images of PAS-stained kidney sections from ADR-treated mice at baseline (0 W), week 2 (2 W) and week 10 (10 W) post-injection: upper panels, low-magnification images (scale bar = 100 μm) and lower panels, high-magnification images of the boxed regions. (**D**–**F**) Quantitative RT-PCR analysis of *Shroom3* expression in the kidney cortex (**D**), isolated glomeruli (**E**) and renal tubules (**F**) from saline- and ADR-treated mice at the indicated time points. Each dot represents an individual mouse datum. Unpaired Student’s *t*-test. * *p* < 0.05.

**Figure 3 cells-14-00895-f003:**
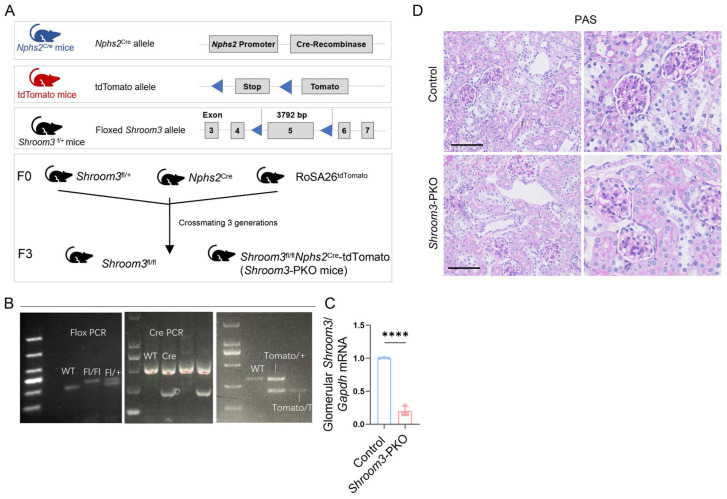
Generation and validation of *Shroom3*-PKO mouse model. (**A**) Schematic representation of the targeting and breeding strategy for generating *Shroom3*-PKO mice. (**B**) Schematic representation of genotyping strategy for mouse identification: left panel provides an illustration of PCR amplification patterns for floxed *Shroom3* alleles (WT: wild-type; Fl/Fl: biallelic flox carriers; and Fl/+: heterozygous flox carrier). Middle panel is a representative diagram showing *Nphs2*-Cre transgene detection. Right panel is an illustrative example of tdTomato reporter integration assessment. Hypothetical DNA ladder markers are depicted on the left of each schematic gel. (**C**) RT-PCR analysis of *Shroom3* expression in isolated glomeruli from control (*n* = 3) and *Shroom3*-PKO (*n* = 3) mice. Unpaired Student’s *t*-test. **** *p* < 0.0001. (**D**) Representative images of PAS-stained kidney sections from 3-month-old control and *Shroom3*-PKO mice. Scale bars = 100 μm.

**Figure 4 cells-14-00895-f004:**
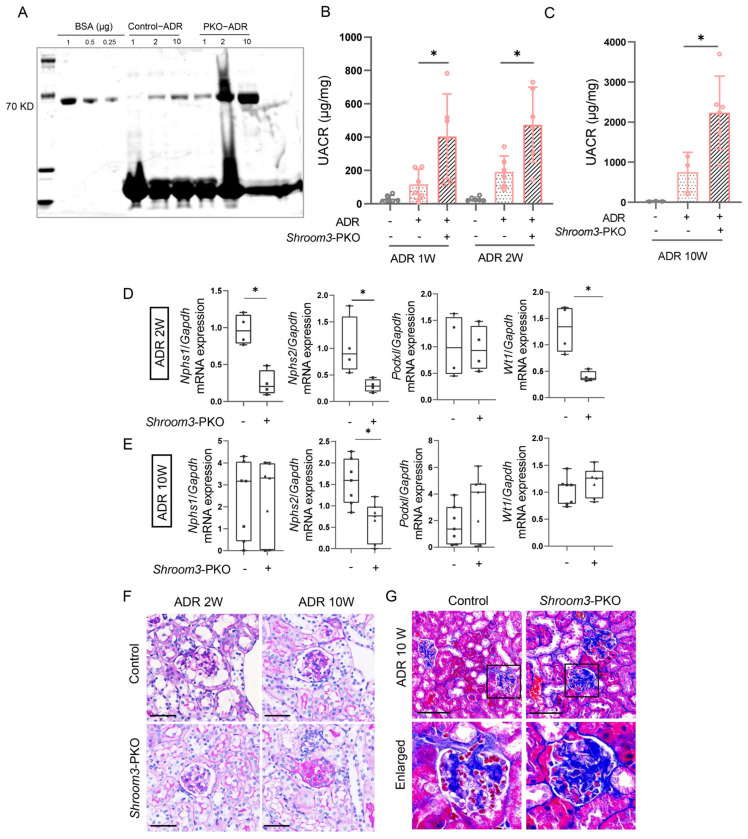
*Shroom3*-PKO mice are more susceptible to glomerular injury induced by ADR. (**A**) Representative SDS-PAGE analysis of urinary proteins from ADR-treated *Shroom3*-PKO and control mice. (**B**,**C**) Quantitative analysis of UACR in ADR-treated *Shroom3*-PKO and control mice at early time points (week 1 and week 2, (**B**)) and a late time point (week 10, (**C**)) post-ADR injection. Each dot represents an individual mouse datum. Unpaired Student’s *t*-test. * *p* < 0.05. (**D**,**E**) RT-PCR analysis of podocyte-specific markers in kidney cortex at week 2 (**D**) and week 10 (**E**) post-ADR treatment. Each dot represents an individual mouse datum. Unpaired Student’s *t*-test. * *p* < 0.05. (**F**) Representative images of PAS-stained kidney sections from control and *Shroom3*-PKO mice at week 2 and week 10 post-ADR administration. Scale bars = 50 μm. (**G**) Representative Masson’s trichrome-stained kidney sections from control and *Shroom3*-PKO mice at week 10 post-ADR treatment. Scale bar = 50 μm.

**Figure 5 cells-14-00895-f005:**
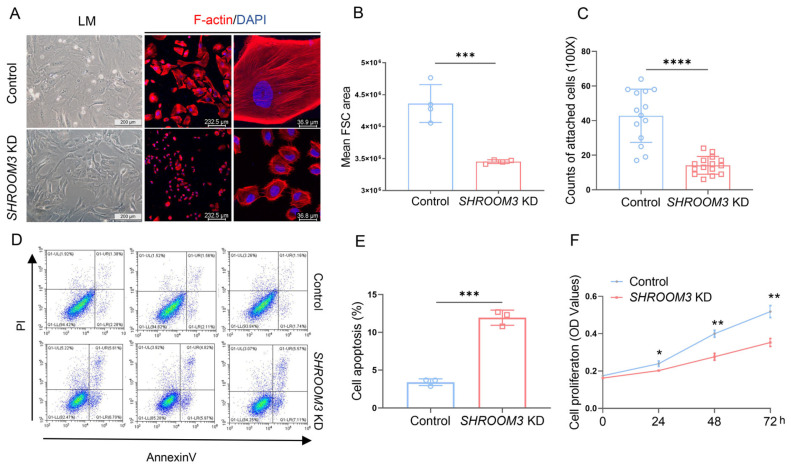
*SHROOM3* depletion alters cytoskeletal architecture, cell adhesion and survival in human podocytes. (**A**) Representative images of control and *SHROOM3*-knockdown (KD) human podocytes: left panels, light microscopy (LM) and middle and right panels, fluorescence microscopy (FM) showing F-actin cytoskeleton (phalloidin, red) and nuclei (DAPI, blue) at low and high magnification. Scale bars = 200 μm (LM), 232.5 μm (FM). (**B**) Quantitative analysis of mean forward scatter (FSC) area from flow cytometry, indicating reduced cell size in *SHROOM3* KD podocytes (*n* = 4). Unpaired Student’s *t*-test. *** *p* <0.001. (**C**) Quantification of attached cells in control (*n* = 14) and *SHROOM3* KD podocytes (*n* = 15) for cell adhesion capacity. Unpaired Student’s *t*-test. **** *p* < 0.0001. (**D**) Flow cytometry dot plots of Annexin V/PI staining for apoptosis detection in proliferated control and *SHROOM3* KD podocytes. (**E**) Quantification of total apoptotic cells (Annexin V+ populations) expressed as a percentage of total cells (*n* = 3). Unpaired Student’s *t*-test. *** *p* < 0.001. (**F**) Cell proliferation of proliferated control and *SHROOM3* KD podocytes assessed by CCK-8 assay over 72 h (*n* = 5 per group). Unpaired Student’s *t*-test. * *p* < 0.05. ** *p* < 0.01.

**Figure 6 cells-14-00895-f006:**
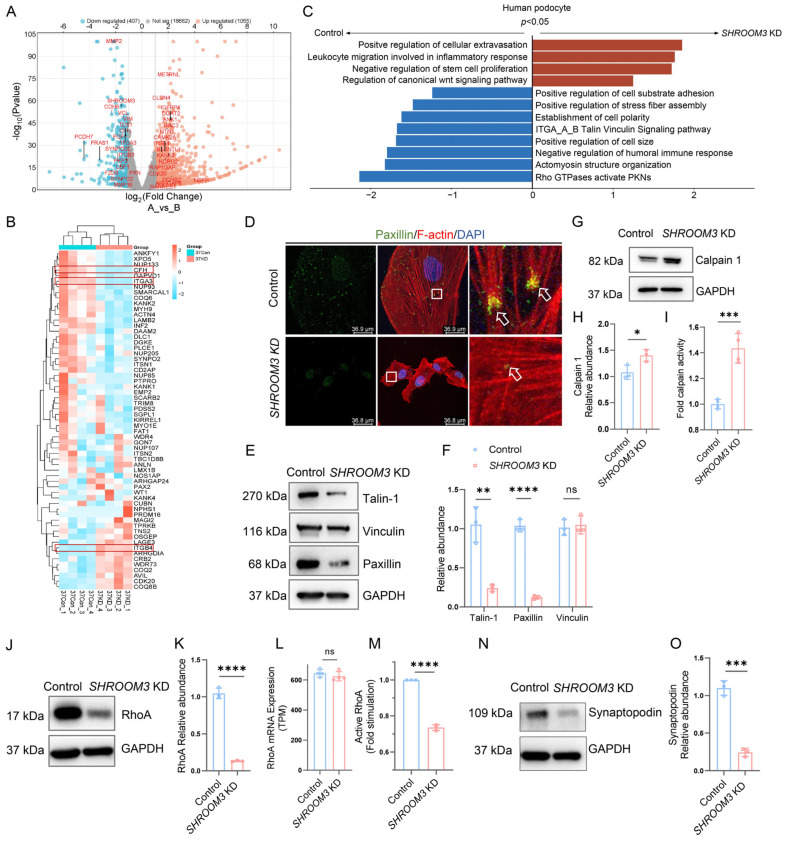
*SHROOM3* deficiency contributes to regulation of podocyte cytoskeletal and focal adhesion dynamics. (**A**) Volcano plot showing DEGs identified by RNA-seq (*SHROOM3* KD vs. control, *n* = 4). (**B**) Hierarchical clustering heatmap of differentially expressed genes from known podocytopathic gene clusters (*n* = 4). Columns represent biological replicates of differentiated control podocytes (37Con) and *SHROOM3* KD podocytes (37KD). (**C**) GSEA revealing significantly enriched pathways in *SHROOM3* KD vs. controls. (**D**) Representative immunofluorescence images showing paxillin (green) distribution and F-actin (phalloidin, red) organization. White arrows indicate areas of colocalization between paxillin and F-actin. Scale bars = 36.9 μm. (**E**) Representative Western blot analysis of focal adhesion protein levels in control and *SHROOM3* KD podocytes. (**F**) Qualification of talin1, vinculin and paxillin protein levels (*n* = 3; Unpaired Student’s *t*-test; ns: *p* > 0.05; **: *p* < 0.01; ****: *p* < 0.0001). (**G**) Representative Western blot analysis of Calpain 1 protein levels. (**H**) Qualification of Calpain 1 protein levels (*n* = 3; Unpaired Student’s *t*-test; *: *p* < 0.05). (**I**) Calpain activity measured by fluorometric assay is increased in *SHROOM3* KD podocytes (*n* = 4; Unpaired Student’s *t*-test; ***: *p* < 0.001). (**J**) Representative Western blot analysis of RhoA protein levels. (**K**) Qualification of RhoA protein levels (*n* = 3; Unpaired Student’s *t*-test; ****: *p* < 0.0001). (**L**) TPM analysis of RhoA mRNA levels from RNA-seq data (*n* = 4; two-tailed Student’s *t*-test; ns: *p* > 0.05). (**M**) RhoA activity measured by GLISA assay is decreased in *SHROOM3* KD podocytes (*n* = 3; Unpaired Student’s *t*-test; ****: *p* < 0.0001). (**N**) Representative Western blot analysis of synaptopodin protein levels. (**O**) Qualification of synaptopodin protein levels (*n* = 3; Unpaired Student’s *t*-test; ***: *p* < 0.001).

**Figure 7 cells-14-00895-f007:**
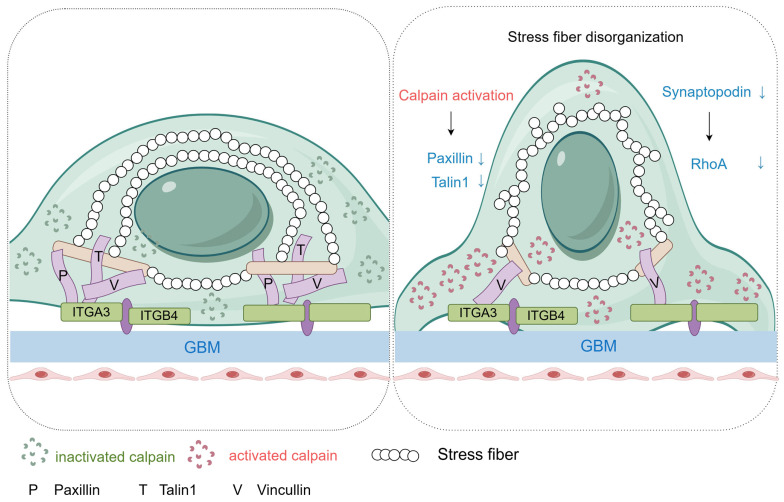
*SHROOM3* deficiency contributes to disassembly of focal adhesions and stress fiber disorganization along with calpain activation and RhoA inactivation. ITGA3, integrin α3; ITGB4, integrin β4; GBM, glomerular basement membrane.

## Data Availability

The single-nucleus RNA sequencing (snRNA-seq) data from healthy adult human and mouse kidneys analyzed in this study are publicly available through the Kidney Interactive Transcript (KIT) database (https://humphreyslab.com/SingleCell/, accessed on 28 February 2025). Transcriptomic data from proteinuric kidney diseases were accessed from the Nephroseq database (https://www.nephroseq.org/resource/login.html, accessed on 28 February 2025), including the “Nakagawa CKD Kidney”, “Mariani Nephrotic Syndrome Glom”, “Mariani Nephrotic Syndrome TubInt2”, and “Hodgin FSGS Glom” datasets. Single-cell RNA sequencing data from the glomeruli of Adriamycin-induced nephropathy mice are available in the Gene Expression Omnibus (GEO) database under accession number GSE146912. RNA sequencing data from human podocyte cell lines generated in this study have been deposited in the GEO database (GSE294734). Additional data supporting the findings of this study are available from the corresponding authors upon reasonable request.

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
