# Peer review of "SHROOM3 Deficiency Aggravates Adriamycin-Induced Nephropathy Accompanied by Focal Adhesion Disassembly and Stress Fiber Disorganization"

_cells, 2025, doi:10.3390/cells14120895_

Round 1
Reviewer 1 Report
Comments and Suggestions for Authors
This study investigated the role of SHROOM3 in proteinuric kidney diseases, with a specific focus on podocyte biology. The authors combine multi, modal approaches—bioinformatics (single-cell RNA-seq and Nephroseq), genetically engineered mouse models, a chemical-induced nephropathy model, and in vitro assays—to demonstrate that SHROOM3 is a crucial regulator of podocyte structure and function. The manuscript mechanisms and suggest potential therapeutic implications. However, a couple of issues in methodology details and mechanistic depth should be addressed to strengthen the manuscript.
Reviewer 2 Report
Comments and Suggestions for Authors
Xu et al submit an original research article entitled "SHROOM3 Deficiency aggravates Adriamycin-induced nephropathy accompanied by Focal Adhesion disassembly and 3 stress fiber disorganization". In this interesting article, they look at a role of SHROOM3 in maintaining podocyte integrity, using a combination of analysis, of already published data, KO of SHROOM3 in an animal model, and immortalized human podocytes.
Concerning RT-qPCR, how was RNA integrity checked? Can the auhtors justify that GAPDH levels did not vary in the different experimental conditions? Concerning the 2-ΔΔCT method, can the authors give more details on the calculation please?
Concerning Western Blot Analysis, the authors do not state which control gene they used.
Concerning podocyte-specific Shroom3 knockout mice, did the authros check their urea and glucose seric levels?
When the authors state "Cre-mediated deletion of exon 5 results in a truncated transcript with a premature stop codon, leading to transcript degradation via nonsense-mediated decay and subsequent absence of protein production." to which data do they refer?
Reviewer 3 Report
Comments and Suggestions for Authors
Authors show a very interesting study, in which through multiple experiments present how SHROOM3 can affect kidney function and what it's its role in kidney damage. Based on in vitro study, then in ADR-induced nephropathy model, and in FSGS patients, Authors indicate that SHROOM3 may be important marker/regulator of acute kidney injury, as well as indicate the rate of kidney damage in FSGS type lesions, when it correlats with GFR.
Comments:
1) I suggest to modify sentence in L39-40, 'nephrotic syndrome' is just a clinical picture of many diseases, but not 'the cause' of CKD, also why Authors mentioned this syndrome in relation to 'first three decades of life'? most CKD patients are older,
2) L115: did you mean ARRIVE? or 1.0 version?
3) Sorry, maybe I missed the information about patients evaluated in the Methods section, and Ethical approval for human experiments/data analysis,
4) in the discussion Authors mentioned '14-3-3' proteins, but did not explain their role in presented study, they were not analyzed here (L552 and later).
Round 2
Reviewer 2 Report
Comments and Suggestions for Authors
changes are ok